# The effect of systemic administration of salicylate on the auditory cortex of guinea pigs

**Mutsumi Kenmochi**◎, **Kentaro Ochi**(iD)*◎, **Hirotsugu Kinoshita**◎, **Yasuhiro Miyamoto**◎, **Izumi Koizuka**◎

Department of Otolaryngology, St. Marianna University School of Medicine, Kawasaki, Japan

◎ These authors contributed equally to this work.
* k2ochi@w6.dion.ne.jp

**Data Availability Statement:** All relevant data are within the manuscript and its Supporting Information files.

## Abstract

### Objective

To investigate the effect of systemic administration of salicylate as a tinnitus inducing drug in the auditory cortex of guinea pigs.

### Methods

Extracellular recording of spikes of the primary auditory cortex and dorsocaudal areas in healthy male albino Hartley guinea pigs was continuously performed (pre- and post-salicylate).

### Results

We recorded 160 single units in the primary auditory cortex from five guinea pigs and 156 single units in the dorsocaudal area from another five guinea pigs. The threshold was significantly elevated after the administration of salicylate in both the primary auditory cortex and dorsocaudal areas. The $Q_{10dB}$ value was significantly increased in the primary auditory cortex, whereas it has significantly decreased in the dorsocaudal area. Spontaneous firing activity was significantly decreased in the primary auditory cortex, whereas it has significantly increased in the dorsocaudal area.

### Conclusion

Salicylate induces significant changes in single units of both stimulated and spontaneous activity in the auditory cortex of guinea pigs. The spontaneous activity changed differently depending on its cortical areas, which may be due to the neural elements that generate tinnitus.

**Funding:** This investigation was supported by a Grant-in-Aid for Scientific Research (C) (JP17591807) to K.O. The funders had no role in study design, data collection and analysis, decision to publish, or preparation of the manuscript.

**Competing interests:** The authors have declared that no competing interests exist.

## Introduction

The administration of large doses of salicylate induces hearing loss and tinnitus in humans. In an animal behavioral study, Jastreboff et al. [1] demonstrated that salicylate induces tinnitus in animals and humans in a similar fashion. Numerous published reports detail the effects of salicylate on various points of the acoustic neural pathway, including the cochlear nucleus [2, 3], inferior colliculus [3–8], and auditory cortex [3, 6, 9–14]. If tinnitus is a neurophysiological product, it may be possible to detect activity changes in the acoustic neural pathway regardless of its cause. Furthermore, because the auditory cortex is the final destination of the acoustic neural pathway, electrophysiological changes in the auditory cortex can be observed regardless of where salicylate acts. In previously conducted animal experiments using sodium salicylate or quinine hydrochloride in the primary auditory cortex of cats [9, 12, 13, 15], we reported a significant increase in the synchronization among all single units without changing the number of spontaneous firings in the primary auditory cortex (AI). Additionally, we reported that salicylate and quinine significantly increased the spontaneous firing rate in the secondary auditory cortex (AII) while reducing it in the AI and anterior auditory field (AAF). We suggested that increased synchronization in the AI and increased spontaneous firing in the AII were the possible neural elements inducing tinnitus.

The abovementioned cortical experiments were conducted mostly using cats. Literature regarding auditory peripheral using cats is scarcer than those using guinea pigs. The guinea pig has been used for auditory research, particularly of the cochlea, because of the easy accessibility to its cochlea. We performed experiments using the guinea pig cochlea, which suggested that salicylate induces changes in the cochlea, such as threshold changes of compound action potential recorded from the round window and decreases in the cochlear blood flow [16].

In contrast, there are only a few reports on the auditory cortex of guinea pigs because the guinea pig is anatomically considered challenging for cortical study. Unlike the cat brain, the guinea pig brain does not have distinct landmarks to recognize the auditory cortex. Wallace et al. [17] reported the cortical map of the auditory cortex of guinea pigs using histological methods with microelectrode mapping. They introduced the following four tonotopically organized areas: the AI, dorsocaudal area (DC), ventrorostral belt, and small field (S). Additionally, there are three belt areas: the dorsorostal (DRB), dorsocaudal (DCB), and ventrocaudal belts (VCB), which surround the DC, S, and part of the AI. In this experiment, we recorded neural activity from the auditory cortex of guinea pigs to compare results with those of some of the previous studies (auditory cortex of cats [9, 12, 13, 15] and cochlea of guinea pigs [16]).

## Materials and methods

### Subjects and surgical procedure

The experiments were conducted on 10 healthy albino Hartley guinea pigs weighing between 355 and 830 g (mean±1 standard error of measurement = 575 ± 61 g).

After subcutaneous injection of 0.1 mL atropine sulfate (0.6 mg/mL), the guinea pigs received an intraperitoneal injection of sodium pentobarbital (32.5–65 mg/mL) at a dose of 25 mg/kg body weight. After 15 min, Xylocaine® (a mixture of lidocaine hydrochloride and epinephrine) was subcutaneously injected for local anesthesia around the trachea, and a tracheostomy was performed. General anesthesia was maintained through tracheostomy by artificial ventilation (2% of sevoflurane and 98% of room air). The depth of anesthesia was monitored by heart rate and the response to body pinch every 30 min. The dosage of sevoflurane was increased by 2% when required. The body temperature of the guinea pigs was monitored using a rectal probe and was maintained at approximately 37 ± 1˚C with a thermostatically

controlled unit (BWT-100, Bio Research Ltd, Nagoya, Japan). After stable anesthesia, the guinea pigs were paralyzed with an intramuscular injection of Relaxin® (suxamethonium chloride) at a dose of 50 mg/kg body weight. The guinea pigs were then placed in a double-walled sound-attenuating room on a vibration isolation frame. After head shaving, Xylocaine® was subcutaneously injected in the right temporal region before the skin overlying the skull was incised. The skin flap was removed, and the skull was cleared of overlying connective tissue. Next, local landmarks on the skull such as the lateral and coronal sutures were verified. Berger et al. [18] reported that the lateral suture forms the border between the parietal and squamosal bones and runs obliquely across the middle of AI, just before turning medially to become the coronal suture. An 8-mm-diameter hole was drilled along the lateral suture wherein its rostal end reached the coronal suture. The hole was occasionally enlarged using small bone rongeurs to expose the pseudosylvian sulcus, thereby allowing complete exposure of the AI and DC. The dura was opened, and the brain was covered with light mineral oil; additional oil was added when required. At the end of the experiments, the guinea pigs were sacrificed with an overdose of sodium pentobarbital.

The maintenance and use of guinea pigs reported in this study were approved (#17591807) by the Life and Environmental Science Animal Care Committee of the St. Marianna University School of Medicine (Kawasaki, Japan).

## Acoustic stimulus presentation

Acoustic stimuli were presented from a loudspeaker (JBL-2450h, JBL Inc, Northridge, California) placed with its center 50 cm in front of the guinea pig's head and perpendicular to an imaginary line through the guinea pig's auditory meatus. The calibration and monitoring of the sound were performed with a condenser microphone (NA-41, Rion Ltd, Tokyo, Japan) positioned above the animal's head.

All stimuli were generated and transferred to the DSP boards of a TDT-2 (Tucker Davis Technologies) sound delivery system. The tone-pip stimulus ensemble comprised three identical sequences of 27 tone bursts covering over 5 octaves (tone separation 1/5 of an octave) from 500 to 17152 Hz presented in pseudo-random frequency order at a fixed intensity level. The characteristic frequencies (CFs) were obtained from the frequency of neural response to the smallest stimulus intensity, and the thresholds were obtained by visual detection of neural responses to the smallest stimulation of sound levels at CFs. The $Q_{10dB}$ value represented the ratio of the CF to the bandwidth 10 dB above the CF and expressed the relative sharpness of frequency tuning.

## Recordings from the auditory cortex

Four shanks (length: 10 mm), each with four microelectrode sites, i.e., 16 microelectrode sites (surface area: 413 $\mu m^2$), with impedances between 1 and 2 MΩ Michigan probe (A4x4-10mm-100-200-413-A16, NeuroNexus, Michigan, USA), were used for extracellular recording of spikes using high-impedance head stages (RA16AC, Tucker Davis Technologies). The microelectrodes were arranged in a 4 × 4 configuration, with a separation of 100 µm between each microelectrode in the array and approximately 200 µm between adjacent shanks. Initially, we used all 16 microelectrodes for recording; however, we sometimes encountered recording difficulties. Thereafter we chose two terminal microelectrodes from each shank (a total of eight microelectrode sites).

The recording was initiated from the point immediately caudal to the pseudosylvian sulcus. This point was reported as the low-frequency units of the AI area [17]. Then, the recording probe was caudally moved to sites with higher frequency units of the AI. The border between

the AI and DC was defined as a line joining the points with a reversal in the tonotopic gradient. Next, the recording probe was further caudally moved beyond this border to sites with lower frequency units of the DC. We planned to record from the AI and DC. Five animals were selected for recording from the AI, whereas the remaining five were selected for recording from the DC. The reference/ground electrode was placed in the neck muscles. The arrays were almost orthogonally oriented toward the cortical surface and manually advanced using a motorized 4-axis micromanipulator (MX7600R, Siskiyou, USA). The depth of recording was between 600 and 900 μm, and the electrodes were likely to be in the deep layer III or IV.

The recorded potentials were amplified $10^4$ times using a pair of amplifiers (RA 16P, Tucker Davis Technologies) and processed using a multichannel data acquisition system (RA 16, Tucker Davis Technologies). The neural activities were recorded using the Brainware software (Tucker Davis Technologies). The neural data were digitized at 12 kHz and band-pass filtered from 300 to 3000 Hz at 9 dB/octave. Spikes were identified online using the trigger levels above the noise level (usually ±14.28 μV). In addition, the artifact rejection was used to extract action potential candidates from the digitized input. Without artifact rejection, any voltage peak exceeding either trigger level will be recorded. With artifact rejection, only candidates of biphasic action potential with both peaks exceeding the respective noise level settings will be recorded. Fig 1 shows an example of spike detection.

Action potentials with both peaks exceeding the respective trigger levels were recorded. Each spike was classified by choosing clusters formed by selecting x- and y-axes among features measured by the Brainware software (Peak 1, Peak 2, Peak to Peak, and Trig to Trig) as shown in Fig 2. We usually choose Peak to Peak or Peak 1 on the x-axis and Peak 2 on the y-axis. We continuously record on the same single unit throughout experimental session. We confirmed waveforms to check single unit separation by choosing clusters.

We usually record four single units per electrode site. Spontaneous activity was based on recordings of 15-min duration without sound while stimulated activity was based on recordings of 15-min duration with sound. It generally took approximately 120 mins from the initiation of the recording to achieving a stable recording and confirming the recording areas (AI or DC). After confirmation of stable recording, stimulated and spontaneous activities were recorded for the baseline pre-salicylate (control) data.

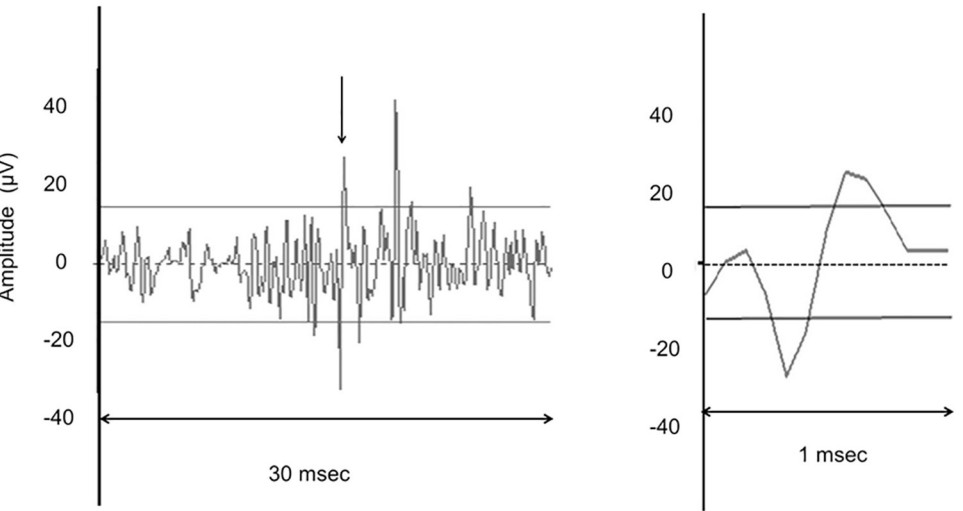

**Fig 1. Example of a spike extraction.** A sample of the 30 ms recording window in the left-hand panel and extended 1 ms recording window in the right-hand panel. Around an arrow (16 ms) in the left-hand panel is extended and an example of the extracted spike is shown (I). The lines represent a trigger level (±14.28μV) setting.

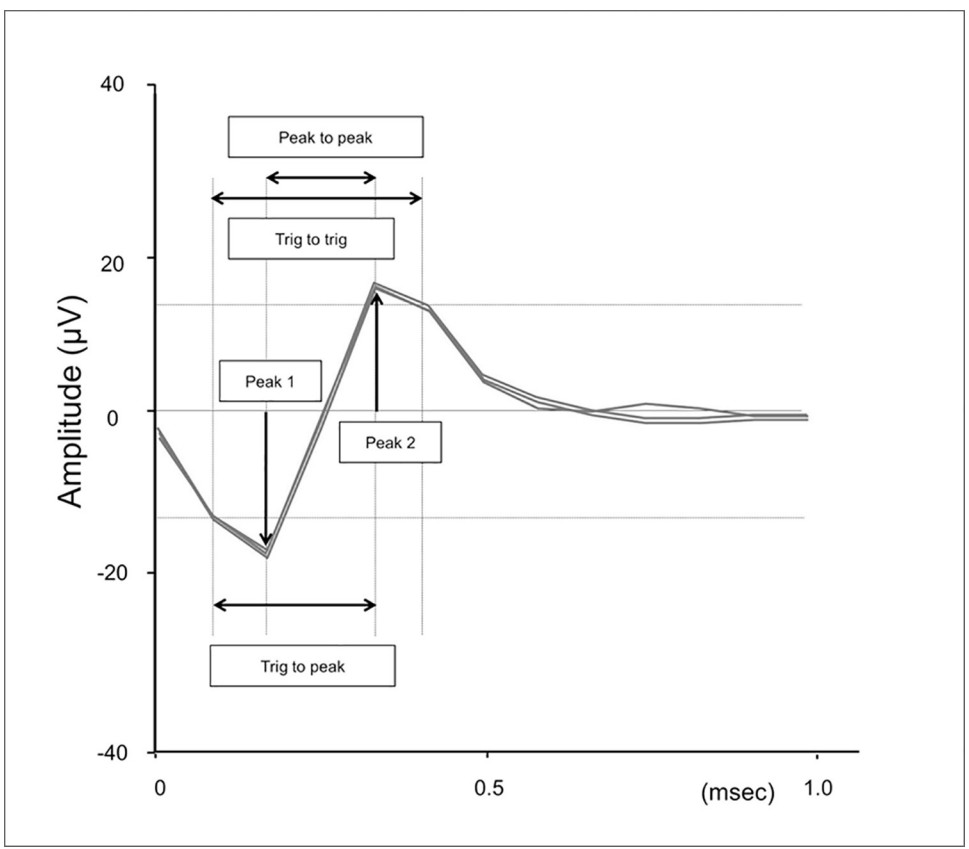

**Fig 2. Features (metrics) measured by the Brainware software.** Three extracted spikes are drawn.

Then, sodium salicylate in a dose of 200 mg/kg (30–60 mg/mL) was slowly administered intraperitoneally. Approximately 30 min after the administration of salicylate, both stimulated and spontaneous activities were recorded for the post-salicylate data. The recording was continued three times at serial time intervals after the administration of salicylate (first session: approximately after 30–60 min; second session: approximately after 60–90 min; third session: approximately after 90–120 min).

## Statistical analysis

All tests were performed using BellCurve for Excel (version 3.21) (Social Survey Research Information Co., Ltd., Tokyo). A $p$-value $< 0.05$ was considered statistically significant. Statistical tests for threshold, CF, $Q_{10dB}$ value, and firing rate between the AI and DC were based on the Mann-Whitney U test, whereas data obtained before and after administration of salicylate were based on the Friedman test. Scheffe's paired comparison was used to compare two groups following the Friedman test.

## Results

### CF, threshold, and $Q_{10dB}$ value

We recorded 160 single units in the AI in five animals and 156 single units in the DC in another five animals. We were not able to record continuously at one electrode site in the DC owing to burst firing. Fig 3 demonstrates the distribution of CFs of the AI and DC. No

(A)

(B)

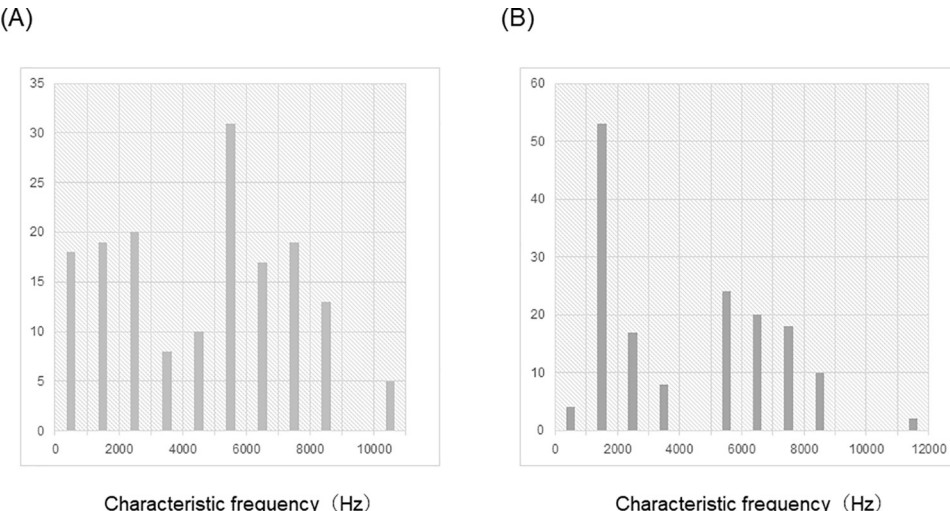

Characteristic frequency（Hz）

Characteristic frequency（Hz）

**Fig 3. Distribution of CF.** The distribution of CFs of the AI and DC are separately shown in (A) AI and (B) DC.

significant difference was noted between CFs of the AI and those of the DC ($p = 0.10$, Mann-Whitney U test).

The thresholds of the AI and DC at before, first, second, and third sessions are separately demonstrated in Fig 4 (A: AI, B: DC). A significant difference ($p < 0.001$) was observed in the threshold of both AI and DC (Friedman test). Scheffe's paired comparison expressed continuous elevation of thresholds in both AI and DC in each session, except for the interval in the DC between before and first session ($p = 0.49$). With respect to threshold elevation, no recovery was observed in this experiment. The thresholds in the AI were significantly ($p < 0.01$) lower than those in the DC (Mann-Whitney U test).

(A)

(B)

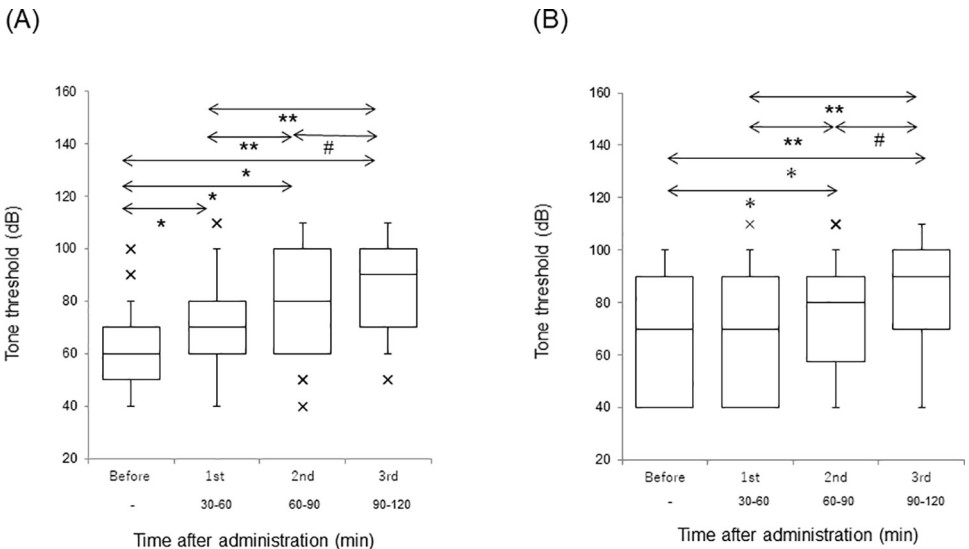

**Fig 4. The threshold at each session.** Each horizontal line indicates tenth, first quartile, median, third quartile, and 90[th] percentile. Outliers are also plotted. (A) AI. *$p < 0.001$ vs. Before; **$p < 0.001$ vs. 1st; #$p < 0.05$ vs. 2nd. (B) DC. *$p < 0.001$ vs. Before; **$p < 0.001$ vs. 1st; #$p < 0.001$ vs. 2nd.

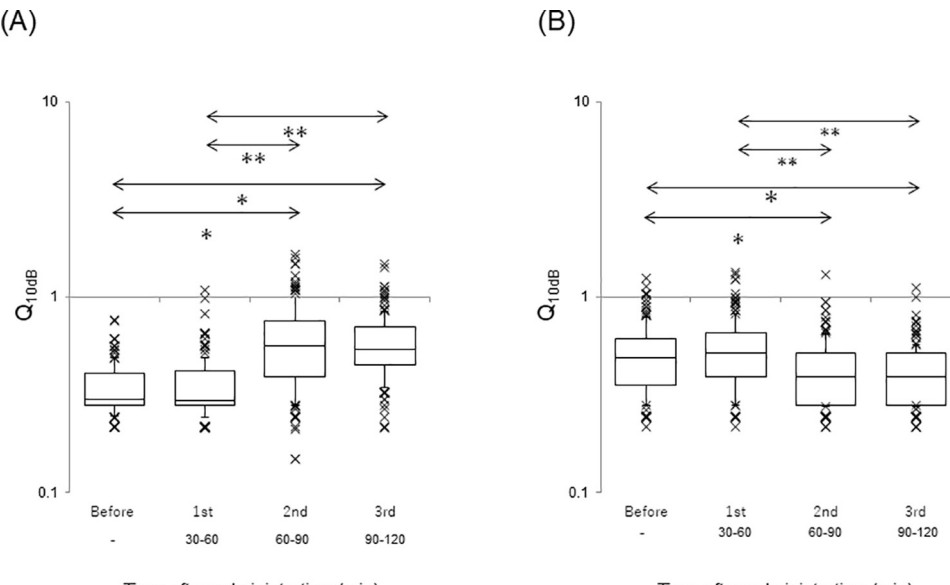

**Fig 5. The $Q_{10dB}$ value at each session.** A logarithmic scale is used. Each horizontal line is the same as Fig 4. Outliers are also plotted. (A) AI. $^*p < 0.001$ vs. Before; $^{**}p < 0.001$ vs.1st. (B) DC. $^*p < 0.05$ vs. Before; $^{**}p < 0.001$ vs. 1st.

The $Q_{10dB}$ values of the AI and DC at before, first, second, and third sessions are separately illustrated in Fig 5 (A: AI, B: DC). A significant difference ($p < 0.001$) was detected in the $Q_{10dB}$ value of both AI and DC (Friedman test). The $Q_{10dB}$ value in the AI of the second and third sessions was significantly increased compared with that of the before session ($p < 0.001$, Scheffe's paired comparison). The $Q_{10dB}$ value in the AI of the second and third sessions was also significantly increased compared with that of the first session ($p < 0.001$, Scheffe's paired comparison). No significant difference was noted in the $Q_{10dB}$ value of the AI in the before versus first session ($p = 0.99$) and second versus third session ($p = 0.92$, Scheffe's paired comparison). In contrast, the $Q_{10dB}$ value in the DC of the second and third sessions was significantly decreased compared with that of the before session (before: $p < 0.05$, Scheffe's paired comparison). Furthermore, the $Q_{10dB}$ value in the DC of the second and third sessions was significantly decreased compared with that of the first session ($p < 0.001$, Scheffe's paired comparison). No significant difference was observed in the $Q_{10dB}$ value of the DC between before versus first session ($p = 0.34$) and second versus third session ($p = 0.99$, Scheffe's paired comparison). The $Q_{10dB}$ value of the before session in the AI was significantly smaller than that of the before session in the DC ($p < 0.001$, Mann-Whitney U test).

## Spontaneous firing

Fig 6 reveals changes in the spontaneous firing rate in the AI and DC. A significant difference was observed in the spontaneous activity of both AI and DC ($p < 0.001$, Friedman test). The spontaneous activities in the AI of the first, second and third sessions were significantly decreased compared with that of the before session (first session: $p < 0.01$; second and third sessions: $p < 0.001$, Scheffe's paired comparison). The spontaneous activities in the AI of the second and third sessions were significantly decreased compared with that of the first session ($p < 0.001$, Scheffe's paired comparison). No significant difference was noted in the spontaneous activity in the AI between the second and third sessions ($p = 0.08$, Scheffe's paired comparison). The spontaneous activity in the DC of the first, second and third sessions were

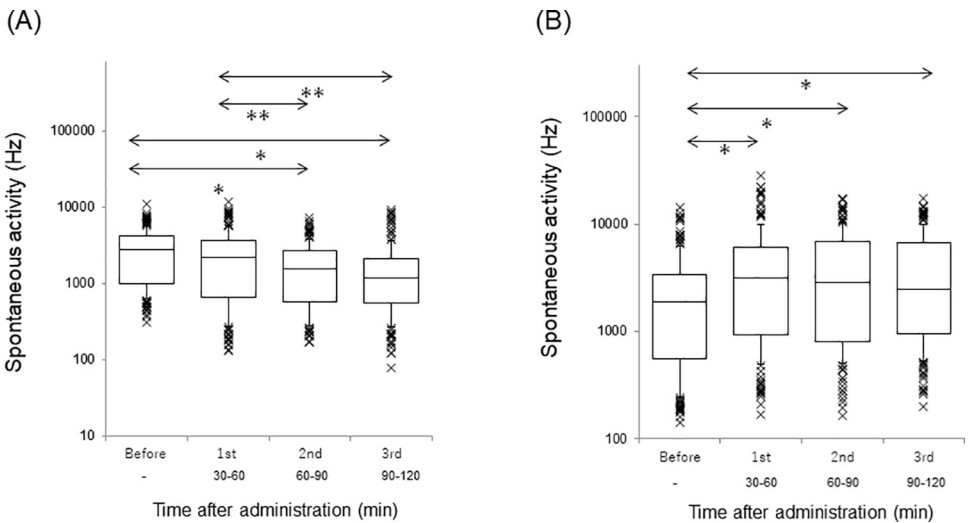

**Fig 6. Spontaneous firing rate at each session.** A logarithmic scale is used. Each horizontal line is the same as Fig 4. Outliers are also plotted. The spontaneous firing rates of the AI and DC at before, first, second, and third sessions are separately revealed. (A) AI. *$p<0.01$, **$p<0.001$ vs. Before; #$p<0.001$ vs. 1st. (B) DC. *$p<0.001$ vs. Before.

significantly increased from that of the before ($p<0.001$, Scheffe's paired comparison). No significant difference was observed in the spontaneous activity in the DC in the first versus second session ($p = 0.99$), first versus third session ($p = 0.75$), and second versus third session ($p = 0.85$, Scheffe's paired comparison). The spontaneous firing of the before session in the AI was significantly higher than that of the before session in the DC (Mann-Whitney U test).

## Discussion

Lower threshold, sharper frequency tuning function and higher spontaneous activity in the AI than that in the DC were recorded before the administration of salicylate. In a previous study, the average threshold in the auditory cortex (AI) of cats underwent a 20–30 dB increase approximately 2 h after the administration of salicylate [13]. We observed similar threshold change in the AI in the second and third sessions, and this was comparable with that of the previous results with respect to the AI of cats.

There is no appropriate method to confirm the presence of tinnitus, and this is considered the main challenge in examining spontaneous activity as a neural element of tinnitus. Correlations between drug concentration and behavioral responses that are indicative of tinnitus have been reported, and several factors that might influence these correlations have been examined [1]. These experiments show that animals may experience tinnitus after the administration of salicylate. Moreover, tinnitus precedes hearing impairment in humans in response to the administration of salicylate [19]. In this study, we observed a significant threshold elevation during the second and third sessions in the AI and throughout the first to third sessions in the DC. In addition, we observed a sharp frequency tuned response in the AI but a broad one in the DC as indicated by the $Q_{10dB}$ value. After systemic administration of salicylate, frequency tuning in the AI was broadened in the first, second, and third sessions, while that of the DC was sharpened in the second and third sessions. These findings suggest that the administration of salicylate induced changes in sound perception at least in the second and third sessions.

Spontaneous firing in the AI was significantly decreased, whereas that in the DC significantly increased by administration of salicylate. The varied effects of salicylate on the current-evoked firing of pyramidal neurons and fast-spiking interneurons in the layer II/III of auditory

cortex slices have been reported in young rats [14]. In the present study, the different changes observed in the auditory cortex may correlate well with the brain slice experiment. We have previously reported increased spontaneous firing in the AII and decreased spontaneous firing in the AI and AAF in the auditory cortex of cats [12]. Compared with our previous cat experiment, the spontaneous activity in the AI of guinea pigs was related to that of the AI and the AAF in cats, whereas the spontaneous activity in the DC of guinea pigs was related to that of the AII in cats. Wallace et al. [17] identified the AI based on its short response latencies, smooth tonotopic gradient, thalamic input from the ventral medial geniculate nucleus, and thick granular structure, which is equivalent to area 41 in other species. Moreover, they mentioned that the AI is the largest auditory field and contains mainly sharply tuned units with an "onset" firing pattern, whereas its intrinsic connections are similar to those of the AI of cats. Summarizing the results from this study and our previous cat study, we assume that the administration of salicylate induces a decrease in the spontaneous activity in the AI of guinea pigs and the AI and AAF of cats and induces an increase in the spontaneous activity in the DC of guinea pigs and the AII of cats.

There are two main possible causes for the changes in the spontaneous firing in the cortex noted in this study. One is the direct effect of the drug on the cortex, and the other is that the drug affects the activity of the neural pathway from the cochlea to the cortex and leads to altered spontaneous firing in the cortex. Salicylate was known to cause functional loss of the outer hair cells while preserving the function of the inner hair cells. We suggest that salicylate-induced tinnitus is possibly initiated by functional changes in the outer hair cells, which may result in a reduction of inhibition at more central levels that induce hypersensitivity and hyperactivity in specific auditory nuclei (probably in the extralemniscal pathway) while reducing the activity in the lemniscal pathway. These suggestions support the increased activity of the DC and decreased activity of the AI in our present study.

The importance of the extralemniscal pathway in the generation of tinnitus has been suggested. The expression of neural plasticity can change the balance between excitation and inhibition in the nervous system and promote hyperactivity; it can also cause reorganization of specific parts of the nervous system or redirection of information to parts of the nervous system that are normally not involved in the processing of sounds (non-classical or extralemniscal pathways) [20].

## Conclusion

In conclusion, the intraperitoneal administration of salicylate induces significant threshold elevation, and the $Q_{10dB}$ value differently changes depending on the auditory area of the guinea pigs: increase in the AI and decrease in DC. Additionally, the intraperitoneal administration of salicylate induces a change of single-unit spontaneous activity in the auditory cortex of guinea pigs: decrease in the AI and increase in the DC. We hypothesize that this change in spontaneous firing in the auditory cortex can be the neural element of tinnitus, but additional behavioral studies are needed.

## Supporting information

**S1 Fig. AI dataset.**
(XLSX)

**S2 Fig. DC dataset.**
(XLSX)

**S3 Fig. AI vs DC dataset.**
(XLSX)

## Acknowledgments

The authors would like to thank the two anonymous reviewers for their helpful comments about the manuscript and Editage for their English language editing services.

## Author Contributions

**Data curation:** Mutsumi Kenmochi, Kentaro Ochi, Hirotsugu Kinoshita, Yasuhiro Miyamoto.

**Formal analysis:** Mutsumi Kenmochi, Kentaro Ochi.

**Funding acquisition:** Kentaro Ochi.

**Investigation:** Mutsumi Kenmochi, Kentaro Ochi, Hirotsugu Kinoshita, Yasuhiro Miyamoto, Izumi Koizuka.

**Methodology:** Mutsumi Kenmochi, Kentaro Ochi.

**Project administration:** Mutsumi Kenmochi, Kentaro Ochi.

**Supervision:** Kentaro Ochi, Izumi Koizuka.

**Writing – original draft:** Kentaro Ochi.

**Writing – review & editing:** Kentaro Ochi.

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
