## [Decision Letter · Decision Letter 0]

1 Jul 2021

PONE-D-21-16756

The effect of salicylate on the auditory cortex of guinea pigs

PLOS ONE

Dear Dr. Ochi,

Thank you for submitting your manuscript to PLOS ONE. After careful consideration, we feel that it has merit but does not fully meet PLOS ONE’s publication criteria as it currently stands. Therefore, we invite you to submit a revised version of the manuscript that addresses the points raised during the review process.

Both reviewers raised concerns about data analysis including single unit sorting, statistical distributions and statistical analysis.  Please also address concerns that the distinction between tinnitus and hearing loss has not been addressed.  Tinnitus can occur in the absence of hearing loss and hearing loss can occur without tinnitus.  Please also address the limitations of using salicylate as a tinnitus inducer.

We look forward to receiving your revised manuscript.

Kind regards,

Susan E Shore, Ph.D.

Academic Editor

PLOS ONE

Journal Requirements:

'The funders had no role in study design, data collection and analysis, decision to

publish, or preparation of the manuscript.'

Additional Editor Comments (if provided):

Reviewers' comments:

Reviewer's Responses to Questions

**Comments to the Author**

1. Is the manuscript technically sound, and do the data support the conclusions?

Reviewer #1: Partly

Reviewer #2: Yes

2. Has the statistical analysis been performed appropriately and rigorously? 

Reviewer #1: Yes

Reviewer #2: No

3. Have the authors made all data underlying the findings in their manuscript fully available?

Reviewer #1: Yes

Reviewer #2: Yes

4. Is the manuscript presented in an intelligible fashion and written in standard English?

Reviewer #1: Yes

Reviewer #2: Yes

5. Review Comments to the Author

Reviewer #1: In “The effect of salicylate on the auditory cortex of guinea pigs”, Kenmochi et al. recorded unit activity in the primacy AC and DC from anesthetized guinea pigs before and after applying salicylate intraperitoneally. They observed a drop in spontaneous activity in primary AC, and a rise in spontaneous activity in DC. Both areas showed an increase in spiking threshold. The authors discuss these findings in the context of tinnitus mechanisms and previous similar work in the cat.

Overall, the work presented appears to be carefully done, but the data presented are very limited in scope and thus this work represents an incremental advance in our understanding of the impact of salicylate on the AC.

There are several suggestions that I have for improving the manuscript.

Major:

1. There is very little analysis provided. After recording from 300+ neurons in the AC, the authors only provide information on thresholds and spontaneous activity. It would be very useful, both from the perspective of comparison to previous work and to better understand the currently-presented findings, if the authors provided additional analyses. For example, the authors measured many units simultaneously, but provide no information about correlations between neurons (though reference previous work about correlations between neurons). There is no information about the impact of CF, of frequency tuning, of depth or any other parameter, in their analysis. The absence of these types of analyses diminish the impact of this manuscript.

2. There is no mention of the impact of the salicylate exposure to the peripheral auditory system/ABRs, etc. Was any analysis of the peripheral system done?

3. Figure 1: Several voltage excursions exceed threshold. Which ones were counted as spikes? Where is the signal taken from that is expanded on the right?

4. Figure 2: Please show histograms of the CFs in the two areas. Seeing their mean/IQRs is not as helpful as just seeing the distributions

5. Figure 3: Please label the y-axis as “Tone threshold (dB)” and please separate AC from DC so that the reader can see the time trends more clearly. Please label significant changes as appropriate.

6. Figure 4: Please label the y-axis as “Spontaneous activity (Hz)” and please separate AC from DC so that the reader can see the time trends more clearly. Please label significant changes as appropriate.

7. The authors make the statement on P15, line 215-216 that the animals were likely to have tinnitus. There is no basis for this statement. Please remove.

Minor:

1. Title and abstract should indicate that salicylate was systemically administered

2. L18 should be “salicylate as a tinnitus-inducing…”

3. L39 should be “regardless of where salicylate acts…”

4. L42 should be “among all single units…”

5. L49 should be “is more scarce…”

6. L55 “unsuitable” is probably not the right word because it implies that this is not an appropriate structure/species to record from. Consider “difficult”, “challenging” or “suboptimal”

7. L141 should be “levels were recorded.”

8. L143 – was only the p-p amplitude recorded? What about spike width or other features? Usually there is an additional dimension that is used for clustering.

9. L250 should be “In conclusion, intraperitoneal salicylate…”

Reviewer #2: In this study, the authors measured the effects of salicylate treatment on the response threshold and spontaneous firing activity in the primary auditory cortex (AI) and dorsocaudal (DC) field of the auditory cortex in guinea pigs. They report that the threshold was increase by salicylate for both AI and DC. Spontaneous activity was suppressed in AI and elevated in DC by salicylate. This study is an extension of their previously published work in which they examine the salicylate effects in Cats. The study was generally well carried out. The findings are interesting and important in understanding salicylate-induced auditory pathologies such as hearing loss and tinnitus. However, there are several points that need to be clarified.

1. The result section and the figure legends are too short and not very clear.

2. The single unit separation was not described in sufficient detail. Was it entirely based on the amplitude of the spikes? Were the same single units tracked over the entire experiment, before and after salicylate application? Were they not confirmed with spike shape?

3. If the 160 and 156 single units were tracked over the four recording periods, the Friedman test (nonparametric version of repeated measure ANOVA) should be used to test the effects of salicylate. It seems that the current statistical tests did not correct for multiple comparison.

4. The graphs plotted top and bottom 10% points as outliers. Are the outliers included in the statistical analysis?

5. The authors used guinea pigs weighing between 355 and 830 g (mean ± 1SEM = 575 ± 61 g). Why are they so different in weight? Is there any age difference? If so, is salicylate effect dependent on the age.

6. In the discussion, the authors argue that threshold increase is evidence of tinnitus. That does not make much sense. The findings that hearing loss happens after tinnitus in humans taking salicylate indicate that hearing loss and tinnitus are two separate effects of salicylate, not causally related.

6. PLOS authors have the option to publish the peer review history of their article (what does this mean?). If published, this will include your full peer review and any attached files.

Reviewer #1: No

Reviewer #2: No

---

## [Author Response · Author response to Decision Letter 0]

27 Sep 2021

Reviewer 1

Thank you for reviewing our manuscript.

We have carefully considered the comments provided by you and the other reviewer, and the manuscript was revised accordingly.

Major

1. We have added the analysis of frequency tuning by measuring the Q10dB value. 

2. We have not recorded the peripheral auditory system in this experiment. In the previous study, we administered salicylate at a dose of 100 mg/kg. In the present study, we applied 200 mg/kg. We believe that the dose is enough for inducing changes.

3. We have changed the figure accordingly.

4. We have added a new figure about the CF distribution of two areas.

5. We have separately added a figure about threshold change with respect to time.

6. We have added a figure about spontaneous activity change with respect to time.

7. Following your comment, we have deleted the sentence.

Minor:

1. We have revised the title and abstract to indicate that salicylate was systemically administered.

2–7. Following your suggestion, we have revised the sentences.

8. We have changed the sentences and added a figure accordingly. 

9. We have changed the sentences as suggested. 

 

Reviewer 2

Thank you for reviewing our manuscript.

We have carefully considered the comments provided by you and the other reviewer, and the manuscript was revised accordingly.

1. We have changed the explanation of the result and legend.

2. We have added a figure and descriptions about single unit separation. We continuously recorded the same single unit throughout the experimental session. We confirmed waveforms to check for single unit separation by choosing clusters. 

3. We have applied the Friedman test for the analysis.

4. Out point was not used for analysis. In addition, it was used not as a numeric value but as an inverted rank.

5. It depends on the animal stock. The animals were 3 weeks old upon arrival. We usually experiment during 2–10 weeks after their arrival. We intended to minimize influences, such as age, animal’s condition, condition of the experimental room, anesthesia, etc.; therefore, we continuously recorded the same single unit throughout the experiment. We mainly applied the paired test for the analysis instead of using a non-paired test.

6. We have deleted the sentences as suggested.

---

## [Decision Letter · Decision Letter 1]

12 Oct 2021

The effect of systemic administration of salicylate on the auditory cortex of guinea pigs

PONE-D-21-16756R1

Dear Dr. Ochi,

We’re pleased to inform you that your manuscript has been judged scientifically suitable for publication and will be formally accepted for publication once it meets all outstanding technical requirements.

Kind regards,

Susan E Shore, Ph.D.

Academic Editor

PLOS ONE

Additional Editor Comments (optional):

Reviewers' comments:

Reviewer's Responses to Questions

**Comments to the Author**

1. If the authors have adequately addressed your comments raised in a previous round of review and you feel that this manuscript is now acceptable for publication, you may indicate that here to bypass the “Comments to the Author” section, enter your conflict of interest statement in the “Confidential to Editor” section, and submit your "Accept" recommendation.

Reviewer #1: All comments have been addressed

Reviewer #2: All comments have been addressed

2. Is the manuscript technically sound, and do the data support the conclusions?

Reviewer #1: Yes

Reviewer #2: Yes

3. Has the statistical analysis been performed appropriately and rigorously? 

Reviewer #1: Yes

Reviewer #2: Yes

4. Have the authors made all data underlying the findings in their manuscript fully available?

Reviewer #1: Yes

Reviewer #2: (No Response)

5. Is the manuscript presented in an intelligible fashion and written in standard English?

Reviewer #1: Yes

Reviewer #2: Yes

6. Review Comments to the Author

Reviewer #1: The added frequency analysis and edits have improved the paper.

Please be sure to label the y-axis in Figure 3

Reviewer #2: (No Response)

7. PLOS authors have the option to publish the peer review history of their article (what does this mean?). If published, this will include your full peer review and any attached files.

Reviewer #1: No

Reviewer #2: No

---

## [Editor Report · Acceptance letter]

27 Oct 2021

PONE-D-21-16756R1 

The effect of systemic administration of salicylate on the auditory cortex of guinea pigs 

Dear Dr. Ochi:

I'm pleased to inform you that your manuscript has been deemed suitable for publication in PLOS ONE. Congratulations! Your manuscript is now with our production department. 

Kind regards, 

on behalf of

Dr. Susan E Shore 

Academic Editor

PLOS ONE